



# Direct and Indirect Application of Univariate and Multivariate Bias Corrections on Heat-stress Indices based on Multi-RCM Simulations

Liying Qiu[1*], Eun-Soon Im[1,2*], Seung-Ki Min[3], Yeon-Hee Kim[3], Dong-Hyun Cha[4], Seok-Woo Shin[4], Joong-Bae Ahn[5], Eun-Chul Chang[6], and Young-Hwa Byun[7]

[1]Department of Civil and Environmental Engineering, The Hong Kong University of Science and Technology, Hong Kong SAR, China
[2]Division of Environment and Sustainability, The Hong Kong University of Science and Technology, Hong Kong SAR, China
[3]Division of Environmental Science and Engineering, Pohang University of Science and Technology, South Korea
[4]Ulsan National Institute of Science and Technology, South Korea
[5]Pusan National University, South Korea
[6]Kongju National University, South Korea
[7]National Institute of Meteorological Science, South Korea

*Correspondence to*: Eun-Soon Im (ceim@ust.hk) and Liying Qiu (liying.qiu@connect.ust.hk)

**Abstract.** Statistical bias correction (BC) is a widely used tool to post-process climate model biases for heat-stress impact studies, which are often based on indices calculated from multiple dependent variables. This study compares five bias correction methods (four univariate and one multivariate) with two applying strategies (direct and indirect) for correcting two heat-stress indices with different dependencies on temperature and relative humidity, using multiple Regional Climate Model simulations over South Korea. It would be helpful for reducing the ambiguity involved in the practical application of BC for climate modeling as well as end-user communities. Our results demonstrate that the multivariate approach can improve the corrected inter-variable dependence and therefore benefit the indirect correction of heat-stress indices that depend on the adjustment of individual components, especially those relying equally on multiple drivers. On the other hand, the direct correction of multivariate indices using the Quantile Delta Mapping univariate approach can also produce a comparable performance in the corrected heat-stress indices. However, our results also indicate that attention should be paid to the non-stationarity of bias brought by climate sensitivity in the modeled data, which may affect the bias-corrected results unsystematically. Careful interpretation of the correction process is required for an accurate heat-stress impact assessment.

## 1 Introduction

Climate models unavoidably produce biased representations of the simulated variables, and it is more problematic not to know how these biases translate into the modeled response to external forcings such as the $CO_2$ concentration, which is known to be responsible for global warming. Therefore, statistical bias correction (BC) of climate model outputs has been progressively adopted as a standard procedure to improve their performance, in particular when feeding them into various climate change impact assessments (e.g., G. Kim et al., 2020; K. B. Kim et al., 2022; Masaki et al., 2015; Qiu et al., 2022; Schwingshackl et





al., 2021). Indeed, the visible benefits archived by adjusting simple statistics (e.g., mean, variance) have led to the wide application of BC. A significant body of research demonstrated that the systematic biases observed in the long-term pattern of the current climate can be well eliminated even when using a very simple technique (e.g., linear scaling). However, the

35 effectiveness of BC methods and their improper assumptions (e.g., statistical stationarity) remain a topic for debate (Maraun et al., 2017). For example, the nonstationary model bias and the large monthly/seasonal correction factor can potentially degrade the BC's performance, particularly with respect to misleading interpretations of extremes (Chen et al., 2021; Lee et al., 2019a, b). On the other hand, the choice of BC approaches in different contexts (e.g., heat-stress impact study, hydrological impact study, adjustment of boundary conditions in downscaling) needs careful assessments case by case (Ehret et al., 2012;

40 K. B. Kim et al., 2022; Rocheta et al., 2017; Zscheischler et al., 2019).

A variety of BC methods with different levels of complexity and performance have been developed and implemented for both global and regional climate simulations (François et al., 2020; Y.-T. Kim et al., 2021; Teutschbein & Seibert, 2012). Generally, their aim is to correct certain features in the target's distribution, such as the simple statistics of the mean (Linear Scaling, LS, Teutschbein & Seibert, 2012) and variance (Variance Scaling, VA, Chen & Dudhia, 2001), or the more advanced quantiles

(Quantile Mapping, QM) for adjusting the entire distribution by parametric (PQM) or empirical (EQM) transformation (Gudmundsson et al., 2012; Switanek et al., 2017). Continuous efforts have also been made to eliminate the drawbacks of existing BC approaches. Quantile Delta Mapping (QDM, Cannon et al., 2015), for example, is designed to explicitly preserve the long-term trend that may be artificially distorted in QM. Nonetheless, all the approaches described above correct bias in a univariate context. They cannot adjust the inter-variable dependencies, which are important for representing physical processes

and estimating compound hazards. It was not until quite recently that the multivariate BC technique was considered and proposed (e.g., Bárdossy & Pegram, 2012; Cannon, 2018; Mehrotra & Sharma, 2015, 2016; Robin et al., 2019; Vrac, 2018), and they have been applied to various climate change impact studies (Dieng et al., 2022; Meyer et al., 2019; Qiu et al., 2022; Zscheischler et al., 2019). Although it is intuitively recognized that multivariate BC could be more suitable for dealing with climate variables characterized by a strong physical linkage in nature, an unambiguous assessment of univariate and

multivariate BC methods is essential to understand the potential limitations of individual methods and to avoid misleading application.

On the other hand, despite the BC method used, when correcting the multivariate indices representing compound hazards, the index can either be directly adjusted using BC techniques, as in the majority of studies (Coffel et al., 2017; Kang et al., 2019; Schwingshackl et al., 2021), or be indirectly corrected that its components are individually corrected prior to the index

calculation (Casanueva et al., 2019; Zscheischler et al., 2019). In this regard, there have been few systematic comparisons of how the direct and indirect use of univariate and multivariate BC methods, respectively, affect the multivariate indices adjustment. Only Casanueva et al. (2018) tested the direct and indirect use of EQM in correcting the multivariate fire danger index, while several studies compared the indirect use of univariate and multivariate BC methods in impact assessments (e.g., Cannon, 2018; François et al., 2020; Zscheischler et al., 2019). Although Casanueva et al. (2018) pointed out that the direct

application of EQM outperforms the indirect one, how it compares with the multivariate BC method remains unknown. Therefore, there is room for a more comprehensive assessment of the effects of univariate and multivariate BC under direct and indirect applying strategies, which may vary along with the dependence structure of the multivariate indices and may affect correction efficiency since the multivariate approach has a higher computation cost.

In this study, we investigate the effects of different BC methods (univariate vs. multivariate) applied with different strategies
(direct vs. indirect) on the statistical adjustment of heat-stress indices that represent the combined effect of human exposure to temperature (T) and relative humidity (RH), using regionally-tailored, fine-scale climate information in Korea from multiple Regional Climate Models (RCMs). The extreme heat is one of the most critical impacts of climate change and we adopt two heat-stress indices with different sensitivities to humidity (Sherwood, 2018), namely, web-bulb globe temperature (WBGT) and apparent temperature (AT). A comparative assessment of the two indices derived from different BC methods and different
strategies will provide valuable insights into understanding how the relationship between heat-stress index and its drivers (e.g., T and RH) affects the performance of univariate and multivariate BC for modeled heat stress. This study will be helpful for reducing the ambiguity involved in the practical application of BC for climate modeling as well as end-user communities.

## 2 Data and Methods

### 2.1 Data

The 3-hourly data used for BC is the near-surface T and RH during the historical period (1979-2014) generated by five RCMs (Table S1) over the CORDEX-East Asia domain (Lee et al., 2020). It is the dynamical downscaling product of the UK Earth System Model (UKESM) in Coupled Model Intercomparison Project Phase 6 (CMIP6). The same variables from ERA5 reanalysis (Hersbach et al., 2018) during the same period are adopted as the observation for BC and validation procedures. For consistency, the variables from all RCMs are first interpolated spatially onto the $0.25° \times 0.25°$ regular latitude-longitude grid
of ERA5 and temporally interpolated onto a standard Gregorian calendar. The analysis focuses only on the land area in South Korea.

### 2.2 Heat-stress Indices

Two popular heat-stress indices are evaluated in this study: WBGT (ACSM, 1984) and AT (Steadman, 1984). There are several different formulations for both indices, and we employ the versions only using T and RH as input variables (Eq. 1-3). Although
both indices are calculated as a function of T and RH, their T/RH dependences are different (Fig. 1). WBGT is more evenly dependent on T and RH, whereas AT relies mostly on T. Also, each index has strengths and limits in evaluating heat-stress impacts (Sherwood 2018; Schwingshackl et al., 2021); thus, they are selected for a more comprehensive evaluation of BC techniques' applicability.

$$WBGT = 0.567T + 0.393e + 3.94 \ (1)$$
$$AT = 0.92T + 0.22e - 1.3 \ (2)$$




T is the near-surface temperature in °C and RH is the near-surface relative humidity in %; $e$ is the vapor pressure (hPa) that can be calculated by

$$e = \left(\frac{RH}{100}\right) \times 6.105 \exp\left(\frac{17.27T}{237.7+T}\right) \quad (3)$$

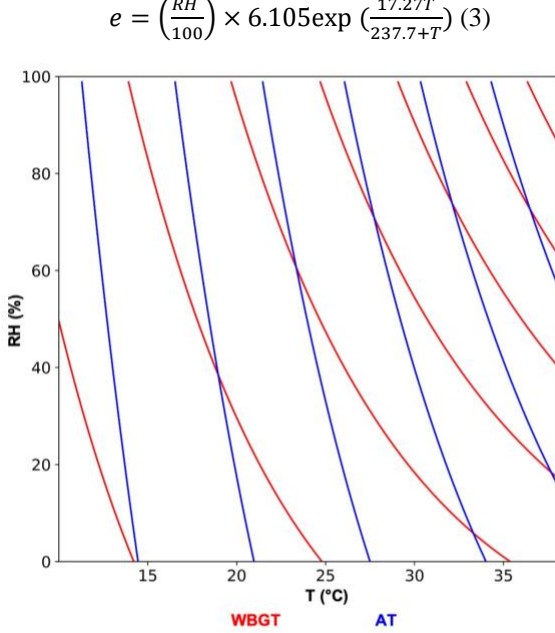

**Figure 1 Contours lines of equal-level heat stress indicators: WBGT (red) and AT (blue)**

All 3-hourly data are used for the BC procedure, but the daily maximum of WBGT/AT during summer (June-July-August, JJA), together with the T and RH at the corresponding time, are selected for analysis in order to facilitate the use of heat-stress impact studies.

**2.3 Bias Correction**

The principle of BC is to use observations to calibrate the simulated output (e.g., climate model output). In this study, five BC methods are applied, including LS, VA, EQM, QDM, and a multivariate BC algorithm with an N-dimensional probability density function (MBCn). Information on each BC approach is provided in the Supplement. The five transformation algorithms cover a varying range of complexity, with MBCn being selected as an example of multivariate correction methods and the trend-preserving QDM being a more "advanced" member of the QM family.

During the BC process, univariate BC methods are applied to T, RH, and WBGT/AT, respectively, after WBGT/AT has been calculated from the original RCM output (ORI). For MBCn, the multivariate approach is applied simultaneously to T, RH, and WBGT (or T, RH, and AT). As the 3-hourly data is adopted, BCs are applied separately to each 3-hour interval in each calendar month (e.g., June00UTC). The direct correction of heat-stress levels is defined as WBGT/AT directly adjusted by BC; on the





other hand, the levels calculated from the bias-corrected T and RH are treated as an indirect correction of the heat-stress indices
(marked as WBGT'/TW'). ENS is the unweighted ensemble mean of the five RCM models.

As an illustrative example, Fig. 2 provides the Quantile-Quantile plots of the WBGT corrected using various approaches for one grid point from one RCM during 1979-1996. ORI shows a cold bias inherited from the driving GCM (M.-K. Kim et al., 2020), leading to a notable underestimation over the entire distribution compared to ERA5. For the direct correction of WBGT, LS reduces the cold bias, but with a non-negligible overestimation, especially in the range of 30-32.5°C. This is due to the
120 asymmetric distribution of T being corrected with a single correction coefficient taken only from the monthly mean. VA, on the other hand, provides a significant improvement by additionally taking the variance into account. EQM and QDM, which are the same in this case for the calibration period, manage to show a perfect match with ERA5 across all the quantiles since the empirical distribution is designed to fit the observation. However, moving to the WBGT' obtained from the corrected T and RH, all univariate BC approaches show a degraded performance while only MBCn retains a qualified correction output.
The MBCn's algorithm ensures that the observed multivariate relations (e.g., the T-RH-WBGT pairwise dependency) are reflected in the corrected distribution, resulting in a better indirect correction outcome.

For cross-validation of the BC methods, we use a historical period of 1979-2014. The first 18 years (1979-1996) are set as the calibration period, during which ERA5 is used to obtain the correcting algorithms, and the second 18 years (1997-2014) are the validation period to which the correcting algorithms from the calibration period are applied. The statistical metrics used
for validation are noted in the Supplement.

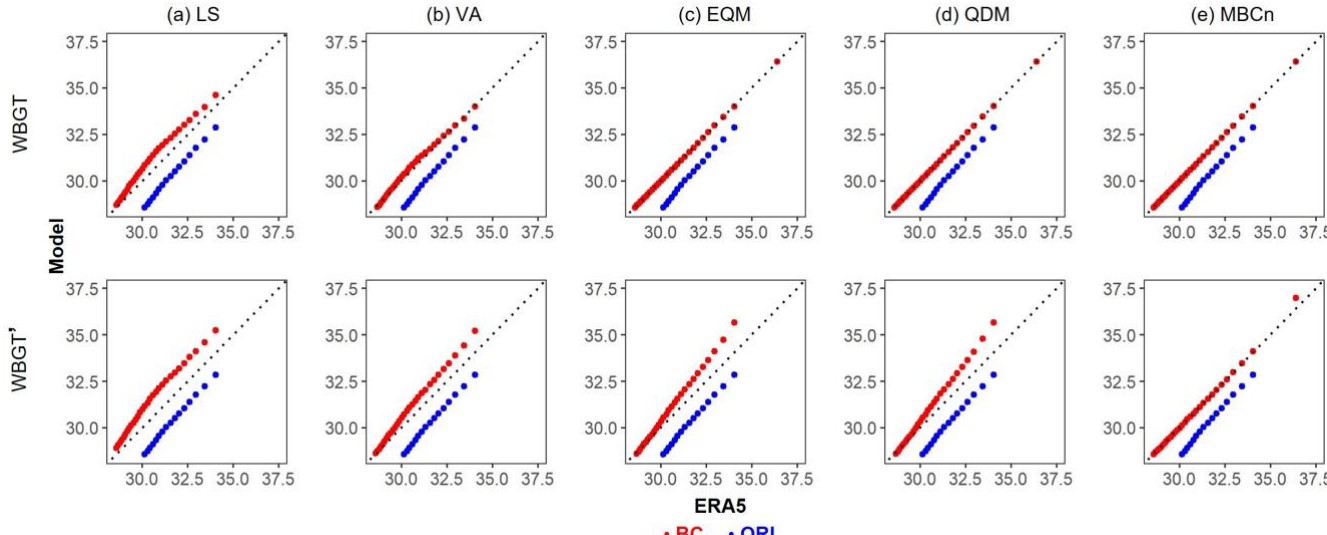

**Figure 2: The Quantile-Quantile plots of ORI (blue) and data after BC (red) adjusted by (a) LS, (b) VA, (c) EQM, (d) QDM, and (e) MBCn. The X-axis is for quantiles from ERA5, and Y-axis is for quantiles from model simulations; the unit is °C. Row 1 is WBGT from direct correction, and Row 2 is WBGT' from indirect correction (calculated from directed T and RH). The data is from one**
**point in GRIMs over South Korea land during the calibration period.**





## 3 Results

Figure 3 presents the performance of WBGT and AT in ORI simulations. Substantial bias can be seen across the entire distribution of the heat-stress indices. For the calibration period, both WBGT and AT generally exhibit a cold bias covering the whole domain. There is more bias in the bottom and top 15% of the distribution, but the bias of WBGT is more skewed to the left tail, whereas that of AT is more skewed to the right. Taking the 90th percentile (90p) as an indicator representing heat

events, Fig. 3b and 3c show a greater cold bias in the low-elevation regions (e.g., basins in southeastern Korea), where an RCM with a spatial resolution of around-20km is highly unlikely to capture the local high temperatures owing to an inadequate representation of topography (Qiu et al., 2020). For the validation period, however, i.e., the next 18 years within the historical period, the cold bias is systematically reduced, with certain area even displaying a slight warm bias. This can be explained by

the high climate sensitivity in the driving GCM (i.e., UKESM; Zelinka et al., 2020), leading to a different level of warming between the simulations and ERA5 during this historical period. According to Fig. 3d and 3e, the model shows around 0.5°C more warming than ERA5 between the two periods, which could in turn "compensate" for the models' cold bias and result in a reduced bias in the validation period. However, while the biased presentation of the heat-stress indices emphasizes the necessity of BC application, the difference in bias between the two historical periods underscores the need for caution when

using and interpreting BC output in climate models, since BC is built on the fundamental assumption of stationary bias (Teutschbein & Seibert, 2012). In particular, the combined bias from climate representation and the long-term trend may amplify the non-stationarity of model biases, thereby causing potential problems in the BC output.

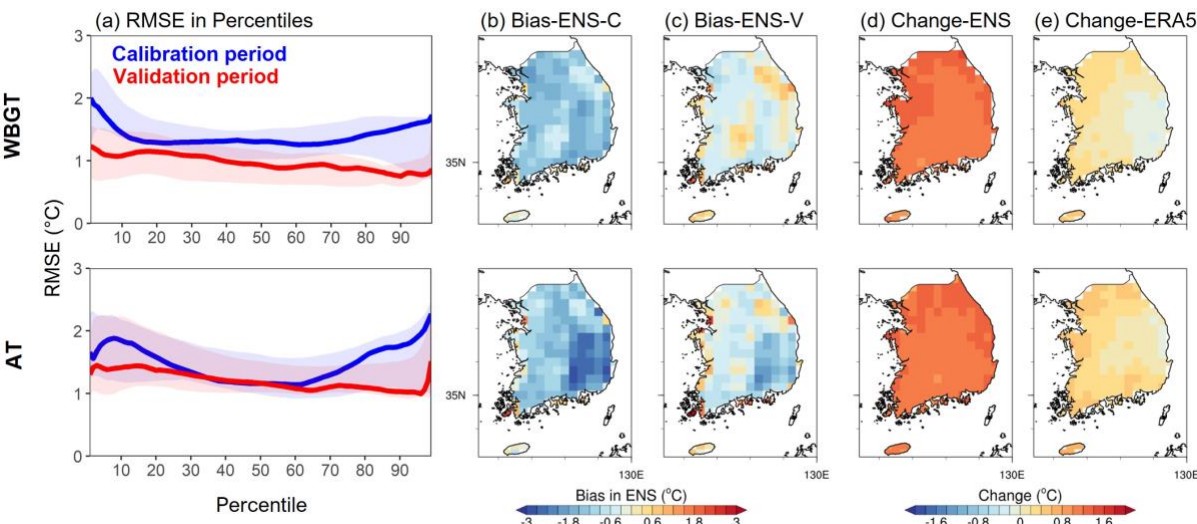

**Figure 3: (a) Root-mean-square-error (RMSE, Eq. B1) over the land area of South Korea in percentiles 1-99 during the calibration**
**(blue) and validation (red) period. The lines and shading indicate the median and the range, respectively, of the five RCMs. (b, c)**
**Spatial map of the bias in the 90p from ENS during the calibration (C) and validation (V) period, respectively. (d, e) The difference**
**between the validation and the calibration period in 90p from ENS and ERA5, respectively. The upper row is for WBGT, and the**
**lower row is for AT.**





Figure 4 shows the median absolute error (MAE, Eq. B2) over South Korea (land only) in all RCMs after BC using different
methods. Two indicators—the 90p and the mean of monthly maximum (MMX)—are selected to represent extreme heat events.
The diamonds standing for ENS are marked for ease of comparison. During the calibration period, LS, as the simplest BC
approach used in this study, shows the largest bias among the five methods. For direct correction of WBGT, all other four
methods have a reasonable MAE of less than 0.25°C in the 90p and less than 0.5°C in the MMX for ENS, with EQM and
QDM slightly outperforming the VA and MBCn approaches. For the indirect correction, on the other hand, there is more
variability among the methods and a larger bias than the direct correction. In this case, while LS still shows the worst
performance, EQM and QDM present a degraded performance, with the MAE for WBGT' reaching 0.6°C and 1.2°C in the
90p and MMX, respectively. Although QDM is supposed to differ from EQM in terms of considering the trend between the
periods, the limited difference between the two historical periods restricts their difference in this study, so we use QDM to
represent the QM family in the below discussion. Surprisingly, VA outperforms the more-advanced QM methods in ENS,
indicating the complexity of using univariate approaches to apply an indirect correction for multivariate hazards. In this case,
the multivariate approach, i.e., MBCn, clearly demonstrates its strengths in such indirect correction, regardless of the indicators
or periods considered. MBCn performs comparably to the direct correction of QDM during the calibration period; however,
for the validation period, MBCn surpasses direct correction with an MAE of roughly 0.5°C for both the 90p and MMX. In
addition, MBCn shows less variability among the RCMs in WBGT'. For example, the range of MAE for WBGT' during the
calibration period as corrected by QDM is 0.38-1.23°C, while that corrected by MBCn is 0.12-0.14°C.

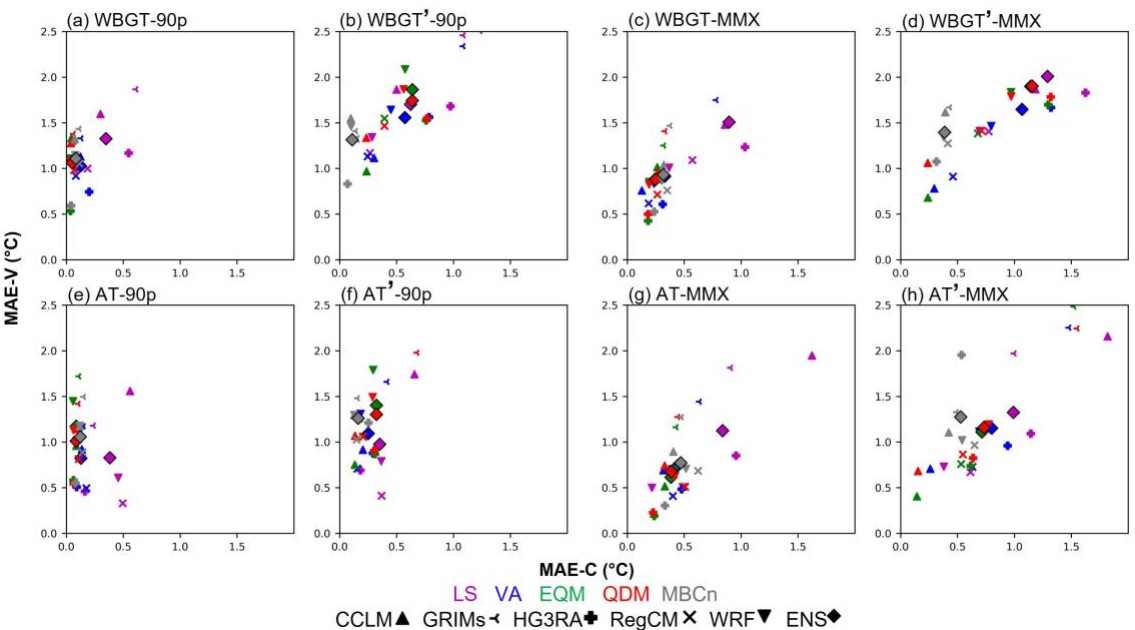

**Figure 4: The MAE over South Korea (land only) for the calibration period (x-axis) and validation period (y-axis) in terms of the (a, b, e, f) 90p, and (c, d, g, h) MMX from (a, c) WBGT, (c, d) WBGT', (e, f) TW, and (g, h) TW'. The different colors stand for different BC methods, and the different markers stand for different RCMs. Some markers are overlapped.**





Similar results are found in AT and AT' according to Fig. 4(e-h). However, for the indirect correction of AT', the weakness
       of QDM is less significant and the advantage of MBCn is also weakened compared to WBGT'. The ability to additionally
       correct the multivariate dependency despite their individual distributions leads to a better result in the indirect correction of
       the heat-stress indices, which are functions of T and RH. In this case, since AT is more reliant than WBGT on T, the effect of
       correcting T-RH interdependence is less critical to its correction outcome. On the other hand, because T and RH both play
important roles in WBGT, multivariate BC is more likely to demonstrate its importance in this case.

       Figure 5 investigates the spatial distribution of bias in the QDM and MBCn corrections, using the 90p as an example for
       WBGT and AT. A similar pattern can also be seen in the case of MMX (Fig. 6). For the calibration period, only indirect
       correction by QDM shows a noticeable warm bias and the bias increases from the northwest to the southeast. Specifically, the
       resultant bias magnitude from indirect QDM correction is even larger than in ORI (Fig. 3b) over southeastern Korea. The
spatial pattern of the warm bias persists in the validation period, although with greater magnitude, which can be explained by
       the different bias magnitudes for the two periods in ORI simulations. This behavior is seen in both WBGT and AT but more
       strongly in WBGT which is more affected by the T-RH dependency. The overall cold bias in the model simulations during the
       calibration period must result in a positive correction coefficient (i.e., towards a warmer condition). However, as discussed
       above, a reduced cold bias in the RCMs is seen in the validation period because of overestimated warming in the models. Such
a "trimmed" bias in the validation period may be over-corrected by the correction coefficient derived from the calibration
       period, even causing a larger bias than in ORI over the eastern part of the country with a warm bias in validation period. Again,
       this warns us of the careful interpretation of bias-corrected climate data, especially in the context of future warming projections.

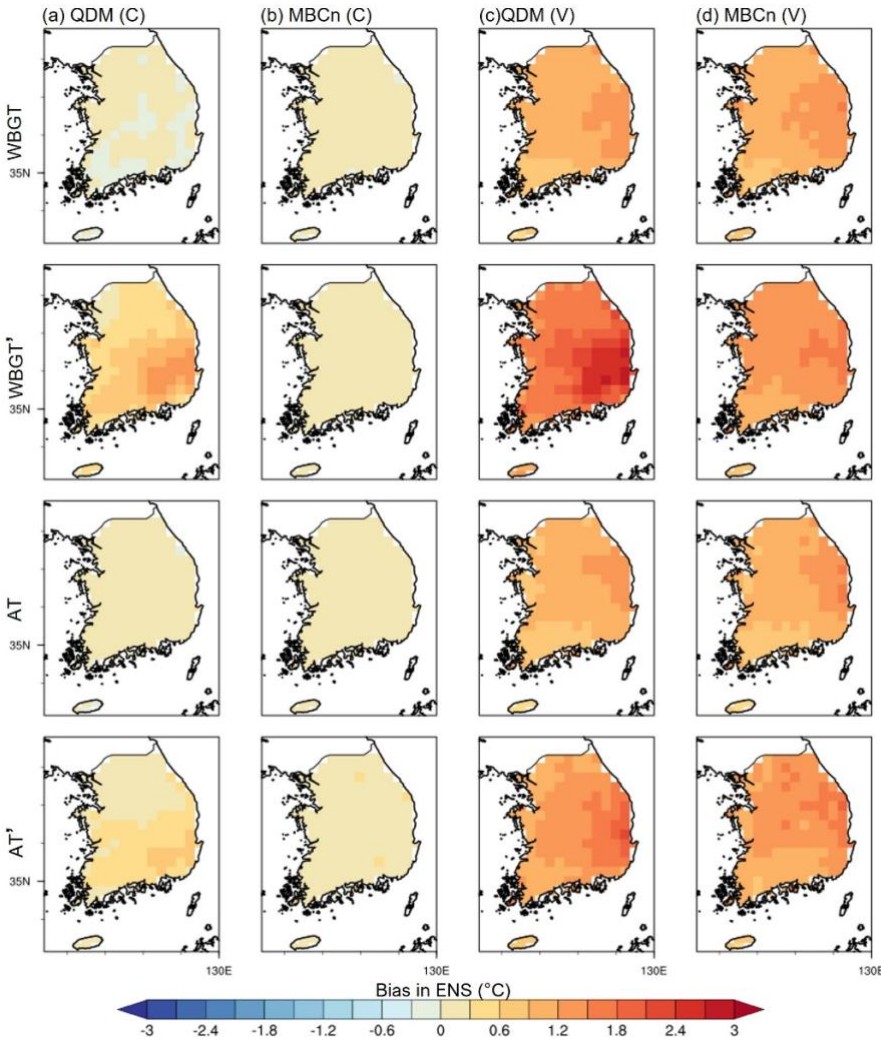

**Figure 5: Spatial maps of the bias in the 90p during the calibration period (C) and validation (V) period corrected by QDM and**
**MBCn in ENS. The first and third rows are the directedly corrected WBGT and AT. The second and fourth rows are the WBGT'**
**and AT' calculated by the corrected T and RH.**

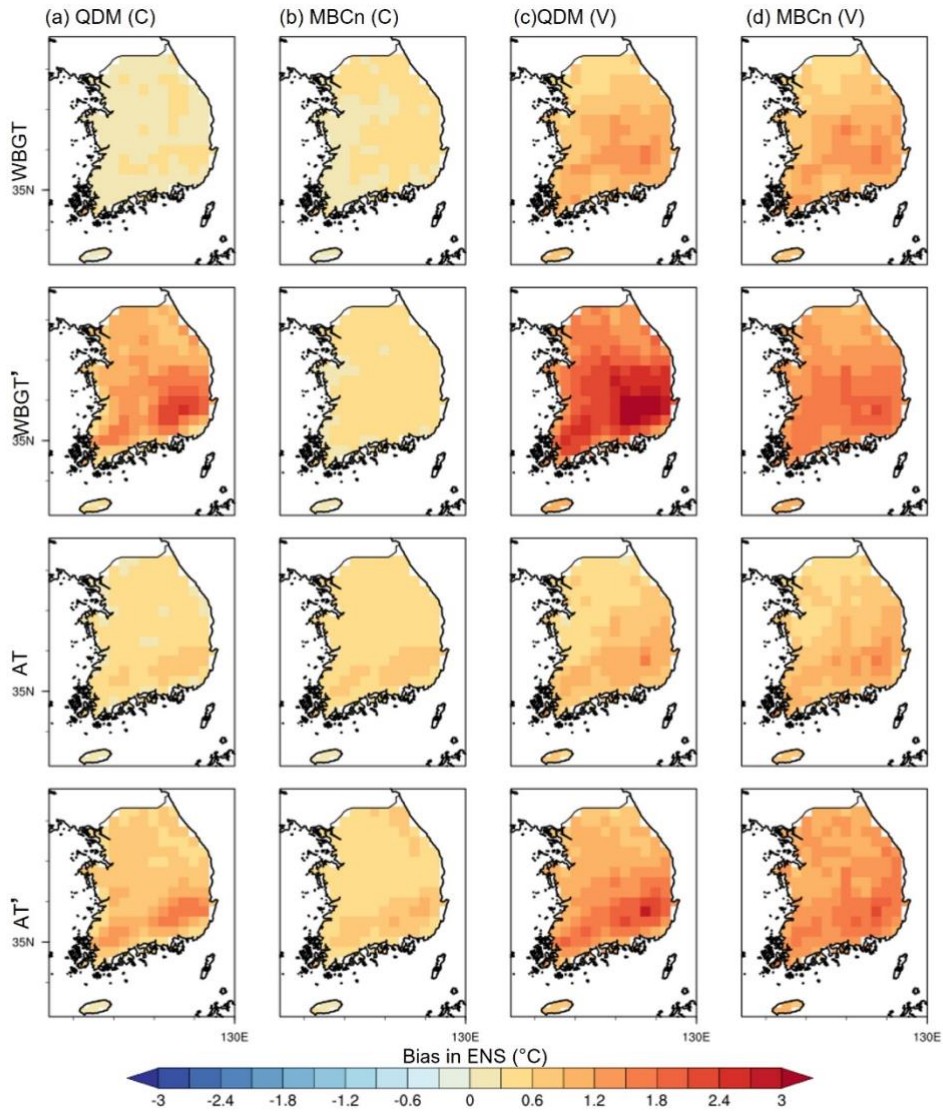

**Figure 6: Same as Fig. 5 but for MMX.**

On the other hand, the spatial maps of bias also clearly demonstrate the superiority of MBCn for the indirect correction of the

205 heat-stress indices over the entire domain in both the calibration and validation periods. Since the heat-stress indices are

functions of T and RH, we investigate the T vs. RH Spearman's rank correlation ($p$-value $< 0.01$) using daily T and RH at the

time when the heat-stress indices reach their daily maxima (Fig. 7). ERA5 shows a negative correlation ranging from -0.4 to -

0.6, that gradually increases from northeast to southwest. Comparatively, ORI has a significantly weaker negative correlation

and does not adequately reflect the spatial gradient. The correction with QDM, even with the good outcome in the direct

correction of the heat-stress indices, cannot properly present the T-RH relation. In fact, it even further weakens their correlation

during the calibration period. On the contrary, MBCn calibrates the multivariate dependency according to the observed





correlation pattern, which explains why it significantly improves the correction of WBGT' and AT'. The correlation derived from the calibration period is also passed to the validation period by MBCn, which in this case shows no significant change between the two historical periods according to ERA5.

**Figure 7: Spatial patterns of T vs. RH Spearman's rank correlation computed in each grid cell during the calibration (rows 1 and 3) and validation (rows 2 and 4) period. Column (a) shows the results from ORI simulations. Columns (b) and (d) are the heat-stress indices directly corrected by QDM and MBCn. Columns (c) and (e) are the heat-stress indices indirectly corrected by QDM and MBCn. Column (f) is from ERA5.**

## 4 Discussion and Conclusion

Previous studies have challenged the applicability of univariate BC for adjusting individual components of multivariate hazard indicators and proved the benefit of multivariate BC in compound event evaluations (François et al., 2020; Zscheischler et al., 2019). Our study also demonstrates MBCn's advantage in correcting the interdependence of the relevant variables, which results in a substantial improvement in the indirect BC of heat-stress indices. Such an advantage is more prominent for the





index relying more equally on the composing variables (e.g., WBGT), which was also pointed out by Zscheischler et al. (2019). However, to the best of our knowledge, no study has been conducted to compare the multivariate BC methods with the direct application of univariate BC on multivariate indices. Our results show that QDM applied directly to the multivariate indices can provide a similar result as MBCn in heat-stress assessments, while MBCn additionally provides a more reasonable underlying inter-variable dependence. In this regard, if only considering heat-stress indices, the more computationally-efficient

QDM direct correction may be sufficient for the impact assessment. On the other hand, if the relationship between T/RH and the heat-related impacts is of interest, the multivariate BC is suggested for maintaining the physical linkage of the variables.

On the other hand, regarding the study of heat stress under future warming that is not evaluated in this study, more aspects should be considered. Our results comparing the validation and calibration period may offer certain insights for future studies: the non-stationarity of bias in the modeled heat-stress indices, as combined effects of internal climate variability and climate

model sensitivity, can significantly affect the BC output. For an appropriate impact assessment, a case-by-case evaluation of BC approaches for certain a climate model and study area, as well as a clear understanding of the relevant processes including the uncertainties underlying original model data, is required.

Meanwhile, for the continuous development in future projections of multivariate heat-stress indices, there are also potential problems worth investigating. For example, we may need to consider if there is any substantial change in the modeled

multivariate dependence structure, which is also highly likely under global warming (Hao et al., 2019; Singh et al., 2021). Although both QDM and MBCn are supposed to preserve the simulated trend in the corrected variables, MBCn, as well as other multivariate BC methods (e.g., $R^2D^2$ developed by Vrac (2018), TSQM developed by Guo et al. (2019)), does not consider the change in the multivariate relationships. In this regard, the direct correction of QDM may outperform MBCn. However, as direct correction of QDM may discard the physical consistency in the input variables, both in terms of the variable

representation and the projected change, it can hide the compensating bias (Schwingshackl et al., 2021) and thus introduce additional uncertainty in climate change signal (Casanueva et al., 2018) in the multivariate heat-stress indices. To solve these problems, a deeper understanding and continuous enhancement in climate models, particularly for the uncertainty and credibility of projections, may be prerequisites for better evaluation and application of the statistical procedures (i.e., BC approaches).

**Code and Data Availability**

Near-surface temperature and relative humidity data from the CORDEX-East domain downscaling product used in this study is archived in the institutional repository at https://doi.org/10.14711/dataset/GTXJVQ. ERA5 hourly data on single levels is downloaded from the Climate Data Store via https://doi.org/10.24381/cds.adbb2d47 (Hersbach et al., 2018). R package "qmap" (https://CRAN.R-project.org/package=qmap, (Gudmundsson, 2016) is used for applying EQM and QDM, and R

package "MBC" (https://CRAN.R-project.org/package=MBC, Cannon, 2020) is used for applying MBCn. Climate Data



Operators (CDO) open-source package is used for 1) computations in LS and VA; 2) temporal and spatial correlation; and 3) statistical analysis.

**Author Contribution**

Im ES and Min SK conceptualized the study and acquired the funding. Qiu LY was responsible for investigation, formal analysis, methodology, software, visualization. Im ES supervised all Qiu LY's work and provided investigations. Qiu LY and Im ES wrote the original draft and Im ES and Min SK reviewed and edited it. Qiu L, Min SK, Kim YH, Cha DH, Shin SW, Ahn JB, Chang EC, and Byun YH created the data used in the study.

**Competing Interests**

The authors declare that they have no conflict of interest.

**Acknowledgments**

This study was supported by the Korea Meteorological Administration Research and Development Program under Grant No. KMI2021-00912.

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
