# Peer review of "Direct and Indirect Application of Univariate and Multivariate Bias Corrections on Heat-stress Indices based on Multi-RCM Simulations"

_Earth System Dynamics, 2022_

## Author Comment (AC1)

**Overall comments**

*This paper describes the differences between ways to achieve bias correction. The outcomes are qualitative in nature so it is difficult to see if they have managed to achieve their aims, as no confidence intervals can be put around the results to examine if they were achieved. For an area that is very keen on numerate approaches I was a little surprised they did not use a statistical approach to differentiate between the differing adjustments for bias.*

► We appreciate the reviewer's helpful comments. In the revised manuscript, we added the Kolmogorov–Smirnov (K-S) test to check the maximum differences between the observed and modelled empirical cumulative distributions and offer a quantitative comparison of the modelled distribution from different BC methods.

Also, for the calculation of Spearman correlation, we set a confidence interval of 99% and the information is added to the text and figure caption in the revised manuscript.

**Section 3**

To assess the quantitative differences in the marginal distributions corrected by different BC methods, Fig. 5 (a , b, e, f) presents the maximum differences calculated from the Kolmogorov–Smirnov (K-S) test (Eq. S3) between the observed (i.e., ERA5) and bias-corrected empirical cumulative distribution functions (CDFs). A smaller value stands for a better correction output. For the direct correction, QDM and MBCn show better performances than LS and VA across all the indices and matrices considered. However, for indirect correction, MBCn shows its unique advantage in the multivariate index depending unequally on the components (i.e., WBGT' in this study), that it can provide a similarly good result in either the direct or indirect correction. In this aspect, QDM shows the largest difference between the direct and indirect applications. Fig. 5 (c, d, g, f) is the D value calculated between outputs from direct and indirect applications of the same BC method, and a smaller value stands for more similar outputs. It clearly indicates a higher similarity seen in the multivariate method than the univariate methods in WBGT, as MBCn successfully retains the intervariable dependence during the correction procedure.

Since the heat-stress indices are functions of T and RH, we investigate the T vs. RH Spearman's rank correlation at a confidence interval of 99%  using daily T and RH at the time when the heat-stress indices reach their daily maxima (Fig. 78).

[Figure]

**Figure 5: K-S test D value between bias-corrected output and observation for (a,e) 90p, and (b,f) MMX, and between direct and indirect corrected output for (c,g) 90p and (d,h) MMX. The D value is ensemble mean of 5 RCMs averaged over South Korea (land only). The different colors stand for different BC methods. The first row is for the Calibration period (C) and the second is for the Validation period (V). In (a, b, e, f), the solid and patterned fill is for the direct and indirect BC, respectively.**

[Figure]

**Figure 8: Spatial patterns of T vs. RH Spearman's rank correlation $(\alpha = 0.01)$ computed in each grid cell during the calibration (rows 1 and 3) and validation (rows 2 and 4) period. Column (a) shows the results from ORI simulations. Columns (b) and (d) are the heat-stress indices directly corrected by QDM and MBCn. Columns (c) and (e) are the heat-stress indices indirectly corrected by QDM and MBCn. Column (f) is from ERA5.**

**Specific Comments**

*The phrase "On the other hand" is used too often.*

► Thank you very much for the comment and we have removed most "on the other hand" and rephrased the corresponding sentences.

**Section 1**

Meanwhile, the choice of BC approaches in different contexts…

Despite the BC method used, …

**Section 2.3**

The direct correction of heat-stress levels is defined as WBGT/AT directly adjusted by BC, while the levels calculated …

**Section 3**

For the indirect correction,  however, there is more variability …

However, if the relationship between T/RH…

---

## Author Comment (AC2)

*The paper compares different bias correction approaches for correcting two heat-stress indices. Although there have been assessments of BC approaches in previous literature, including univariate and multivariate ones, the authors do offer a new perspective of comparing the direct and indirect implementations, which is often confusing for impact studies and thus worth investigating. In this regard, I believe this paper can provide useful information for the community, especially for those processing data on heatwaves or other similar compound indices. However, I have several concerns that should be addressed before the publication:*

► We appreciate the reviewer's positive feedback and helpful comments. Please kindly find our detailed responses to each comment below.

*Major comments*

1. *Section 2.3: The authors selected four univariate BC methods but only one multivariate method (i.e., MBCn) in this paper. Since several different MBC methods have been developed in recent years (e.g., R2D2 (Vrac 2018), MRec (Bárdossy and Pegram), the authors may need to explain why they select MBCn here and what its characteristics are, either in the Introduction or the Method section.*

*Also, although you have included a detailed description of MBCn in the Supplementary Information, I suggest including general information for describing how MBC works (maybe one or two sentences) in the main text for those unfamiliar with MBC.*

► Yes, we recognized that there are different MBC methods available. The key reason we select MBCn here is to facilitate a comparison with QDM, which is the key univariate method used in this study, since MBCn originally stemmed from QDM but with a multivariate structure. In this regard, we made a straightforward comparison only for the multivariate design instead of other assumptions lying in the correction of individual variables. We have added this explanation and a general description of MBC in the manuscript.

**Section 2.3**

> The five transformation algorithms cover a varying range of complexity, with MBCn being selected as an example of multivariate correction methods and the trend-preserving QDM being a more "advanced" member of the QM family. There are several different multivariate BC methods developed recently based on different statistical techniques and/or assumptions (e.g., Rank Resampling For Distributions and Dependences (R2D2, Vrac, 2018), Matrix recorrelation (MRec, Bárdossy and Pegram, 2012)). Different multivariate methods have their own pros and cons, depending on the varying perspectives considered (François et al., 2020). The MBCn adopted here is based on an image processing algorithm that repeatedly rotate the multivariate matrices and apply QDM correction on individual variables, until the multivariate distribution is matched to observation. It is selected in this study not only due to its wide application in various kinds of climate studies; but more importantly, it facilitates the comparison with the univariate QDM as it is built on the latter.

2. *Figure 3: As the bias shown in the calibration and validation periods is different, the authors may consider applying the same experiments with two periods switched to see if the same systematic bias retain and how it affects the bias-corrected result and whether the bias correction model changes significantly. Especially since the authors do not present future projections, using a reverse-periods experiment can increase the robustness of the result.*

► Thank you for the reviewer's comment, and we agree that adopting a reverse-period test can increase the robustness of the result and offer more space for discussion. Therefore, we now include the reverse that the 1997-2014 period is used for calibration and 1979-1996 for validation. The results from the reverse test are also included in the manuscript and Supplement. Also, we recognized that the original Figure 6 provides similar information as the original Figure 5, so we moved it to the Supplement (Fig. S3) and replaced it with the current Figure 7 that comes from the reverse test.

**Section 2.3**

> For cross-validation of the BC methods, we use a historical period of 1979-2014 and adopt the "jack-knifing" split-sample test, that first splits the historical period into two halves and uses one part for calibration and the other for validation, and then reverse the two parts systematically (Refsgaard et al. 2014). Specifically, the 18-year period of 1979-1996 is first set as the calibration part with the period of 1997-2014 as the validation part, then the periods are swapped using 1997-2014 for calibration and 1979-1996 for validation. For each test, the ERA5 data in corresponding calibration period is used to obtain the correcting algorithms that are then applied to the validation period. To distinguish the two tests, the one using 1997-2014 for calibration is all marked with a letter "r" standing for "reverse" and the default is the one using 1979-1996 for calibration. The statistical metrics used for evaluation are noted in the Supplement.

**Section 3**

> …multivariate BC is more likely to demonstrate its importance in this case. Not surprisingly, the performances of different BC methods are retained in the reverse test, although with different magnitudes of MAE (Figure S1). MBCn shows an even better performance in this case, outperforming all other methods despite the heat indices and matrices considered.

> …, even causing a larger bias than in ORI over the eastern part of the country with a warm bias in validation period. The results from the reverse test (Fig. 7 and Fig. S4) can further prove the impact of non-stationary bias on the result. In this case, the validation period of 1979-1996 retains a cold bias after BC for the reason that the correction coefficient derived in 1997-2014 is not large enough to compensate its negative bias. Again, this warns us of the careful interpretation of bias-corrected climate data, especially in the context of future warming projections.

[Figure]

**Figure 7: Same as Fig. 6 but for the reverse test.**

[Figure]

**Figure S1: The MAE over South Korea (land only) for the calibration period (1997-2014, x-axis) and validation period (1979-1996, y-axis) in terms of the (a, b, e, f) 90p, and (c, d, g, h) MMX from (a, c) WBGT, (c, d) WBGT', (e, f) TW, and (g, h) TW'. The different colors stand for different BC methods, and the different markers stand for different RCMs. This result is from the reverse test.**

[Figure]

**Figure S2: K-S test D value between bias-corrected output and observation for (a,e) 90p, and (b,f) MMX, and between direct and indirect corrected output for (c,g) 90p and (d,h) MMX. The D value is ensemble mean of 5 RCMs averaged over South Korea (land only). The different colors stand for different BC methods. The first row is for the Calibration period (C) and the second is for the Validation period. In (a, b, e, f), the solid and patterned fill is for the direct and indirect BC, respectively. This result is from the reverse test.**

[Figure]

**Figure S3: Spatial maps of the bias in the MMX during the calibration period (C) and validation (V) period corrected by QDM and MBCn in ENS. The first and third rows are the directedly corrected WBGT and AT. The second and fourth rows are the WBGT' and AT' calculated by the corrected T and RH.**

[Figure]

**Figure S4: Same as Fig. S3 but for the reverse test**

3. *I am not sure how the authors could solve the problems with non-stationarity with the results of this study, which is indeed a problem of all bias correction. I suggest a discussion with a reverse-period experiment (Comment 2) to emphasize the problem in non-stationary bias, while the authors rephrase the argument with a "softer" tone.*

► Yes, the purpose of this study is not to solve the problem of non-stationarity. Using historical data comprising non-stationarity combined with two split-sample tests, we try to show how non-stationarity affects the output of BC, which is often ignored but important in a warming climate. We have added more discussion regarding this problem in the manuscript based on the new reverse test and literature review.

**Section 4**

>  This study uses historical climate simulations comprising non-stationarity combined with two "jack-knifing" split-sample tests. It

> is found that may offer certain insights for future studies: the non-stationarity of bias in the modeled heat-stress indices, as combined effects of internal climate variability and climate model sensitivity, can significantly affect the BC output. Teutschbein & Seibert (2012) once suggested that the more advanced correction methods (e.g., QM) are more robust to a non-stationary bias compared to the simpler ones (e.g., LS), but our result shows no significant difference. In fact, lying under the fundamental assumption of stationary bias, current BC approaches may not be able to provide a suitable solution to this issue. Therefore,  a case-by-case evaluation of BC approaches for certain aa certain climate model and study area, as well as a clear understanding of the relevant processes including the uncertainties underlying original model data, is required for reliable data post-processing using BC methods.

**Minor Comments:**

1. *P3, Line 88: Instead of "WBGT", the equation (3) used in this paper should refer to "simplified WBGT". The authors should specify this.*

► Thank you for the comment and we have added the explanation about the versions of WBGT and AT we are using in this manuscript.

**Section 2.2**

> Two popular heat-stress indices are evaluated in this study: WBGT (ACSM, 1984) and AT (Steadman, 1984). There are several different formulations for both indices, and we employ the versions only using T and RH as input variables (i.e., the simplified WBGT and the AT without wind effect, Eq. 1-3).

2. *P7, Line 167-170: As you find almost no difference between the results of EQM and QDM in this study due to the use of only historical data, how about keeping just one of these two methods? I feel it redundant to present both here.*

► Thank you for the comment, and we have removed EQM from the manuscript accordingly.

3. *P8, Line 188: I think that this statement is not fully supported by the calibration period, but it's true for the validation period. Therefore, it's better to change the location of this sentence in the paragraph.*

► Thank you for the comment, and we have rephrased this sentence to more precisely describe the figure.

**Section 3**